# Carbon Nanotube-Based Scaffolds for Cardiac Tissue Engineering—Systematic Review and Narrative Synthesis

**DOI:** 10.3390/bioengineering8060080

**Published:** 2021-06-09

**Authors:** Louie Scott, Izabela Jurewicz, Kamalan Jeevaratnam, Rebecca Lewis

**Affiliations:** 1School of Veterinary Medicine, University of Surrey, Guildford, Surrey GU2 7AL, UK; l.w.scott@surrey.ac.uk (L.S.); drkamalanjeeva@gmail.com (K.J.); 2Department of Physics, University of Surrey, Guildford, Surrey GU2 7XH, UK; izabela.jurewicz@surrey.ac.uk

**Keywords:** carbon nanotube, tissue engineering, biomaterial, conductive scaffold, cardiomyocyte

## Abstract

Cardiovascular disease is currently the top global cause of death, however, research into new therapies is in decline. Tissue engineering is a solution to this crisis and in combination with the use of carbon nanotubes (CNTs), which have drawn recent attention as a biomaterial, could facilitate the development of more dynamic and complex in vitro models. CNTs’ electrical conductivity and dimensional similarity to cardiac extracellular proteins provide a unique opportunity to deliver scaffolds with stimuli that mimic the native cardiac microenvironment in vitro more effectively. This systematic review aims to evaluate the use and efficacy of CNTs for cardiac tissue scaffolds and was conducted according to the Preferred Reporting Items for Systematic Reviews and Meta-Analysis (PRISMA) guidelines. Three databases were searched: PubMed, Scopus, and Web of Science. Papers resulting from these searches were then subjected to analysis against pre-determined inclusion and quality appraisal criteria. From 249 results, 27 manuscripts met the criteria and were included in this review. Neonatal rat cardiomyocytes were most commonly used in the experiments, with multi-walled CNTs being most common in tissue scaffolds. Immunofluorescence was the experimental technique most frequently used, which was employed for the staining of cardiac-specific proteins relating to contractile and electrophysiological function.

## 1. Introduction

Cardiovascular disease (CVD) encompasses all illnesses relating to the heart and circulatory system whether genetic, congenital, or environmental. For more than 15 years, it has been the number one cause of death globally, with approximately 80% of CVD deaths occurring in low to middle income countries [1,2]. CVD, therefore, represents a cumbersome burden for health care organizations worldwide, with the United Kingdom alone spending approximately £9 billion per year on CVD health care. This does not include the economic impact due to premature deaths, disability, informal care, and other costs estimated to be £19 billion per year [3]. Despite these acknowledgements, research and development into CVD has been in decline over past decades while research into other diseases like cancer and infectious diseases have continued to excel in line with broader corporate and technological expansion [4].

More recently, cardiac tissue engineering has been heralded by many as the salvation and future of research and development into new treatments for CVD as it provides useful platforms to expedite research in this area [5]. The growth of cardiovascular cell and tissue types on synthetic culture scaffolds has applications both surgically and pharmaceutically as a method of growing transplantative material for diseased and damaged tissues and in biologically representative models for pre-clinical pharmacological testing [6,7,8,9,10]. A key aspect of tissue engineering is the optimization of the structures upon which the tissues are grown, frequently referred to as biomaterials—a moniker derived from the material’s biocompatibility and its interaction with biological systems rather than its composition. Choices by researchers on what features to employ in a biomaterial scaffold are often dictated by the demands of the cell type being used. In the case of cardiac cell types, these demands can be very particular and must be adhered to for success [11]. Currently, there is no biomaterial that can precisely reproduce the highly complex nature of the native cardiac extracellular matrix (ECM) environment. Hence, such approaches and protocols for cardiomyocytes (CMs), to date, are yet to be perfected.

CMs are a highly specialised cell type with a decidedly recognisable rectangular ‘rod’ shape. They range from 10–25 µm in diameter and approximately 100 µm in length with an aspect ratio between 5 and 7 to 1 [12,13].They are able to adhere to substrates through membrane surface proteins such as talin, vinculin, and integrins [14]. In vitro CMs will contract through internal structures known as sarcomeres and will synchronise contractions with adjacent CMs via electrical junctions contained within intercalated disk structures [15,16]. However, maintenance of these cells, or derivatives such as CMs created from human induce pluripotent stem cells, in vitro for extended periods of time is currently challenging and remains somewhat of an art [17]. While there are instances of CMs creating representative engineered tissues when co-cultured with other cardiac cell types, cells in vitro frequently fail to emulate the behaviour of cardiac muscle *in vivo* [10,12,18,19]. While many issues can be attributed to this failure, one of the most important from the perspective of materials engineering is the lack of suitable tissue scaffolds that satisfy the complex combination of stimuli required by cardiac cell types. The specialisation of cardiac muscle is in part due to the highly adapted environment in which it operates. This calls for more advanced tissue scaffolds that can satisfy the requirements of the CMs by effectively mimicking the native myocardium.

Carbon nanotubes (CNTs) have been identified as a promising biomaterial for use in tissue engineering in general [20,21,22,23], but their unique properties are particularly appealing for cardiac tissue engineering [24]. A CNT is a one-dimensional (1D) allotrope of carbon composed of sheets of graphene rolled into hollow tubes with diameters in the nanoscale and lengths generally in the microscale [25]. CNTs can be divided into single-walled (SWNTs), double-walled (DWNTs), and multi-walled (MWNTs) [26]. Following their discovery in the late-20th century [27], CNTs have revealed the remarkable properties that has made them an ideal candidate for numerous applications including electronics and optoelectronics, chemical and mechanical sensors, mechanical actuators, and composite materials [28,29,30,31].

In terms of mechanical properties, CNTs have surpassed all pre-existing materials previously considered to be the strongest and stiffest ever discovered with tensile strength up to 1 TPa and Young’s modulus up to 100 GPa [32,33,34,35,36]. CNTs also display exceptional electrical and thermal conductivity that can be modulated by changing various features such as the number of walls, the chirality of bonds, and tube diameter to name a few [26,37,38]. Despite all their positive attributes, CNTs’ inert nature, due to being solely composed of carbon, restricts their applicability in many situations. CNTs are frequently produced and purchased as powders, so converting this raw product into functional liquid form is an easy, cheap, and versatile way to permit the production of larger scale devices, functional materials, and synthetic scaffolds. Amide solvents such as *N,N*-dimethylformamide (DMF) and *N*-methylpyrrolidone (NMP) are frequently used to produce CNT dispersions, [39,40], but even in this case, the surface energy of CNTs can significantly differ from the organic solvents or polymer matrices [41], thus, making ease of processability a major obstacle when attempting to utilise CNTs. However, dispersibility of CNTs in aqueous media is a fundamental prerequisite for their bio-applications. CNTs can be covalently functionalised by the addition of chemical moieties, [42], or non-covalently functionalised through the use of surfactants [43,44]. Considerations must be made at every step of production as, for example with biological applications, high boiling point organic solvents and surfactants are highly cytotoxic and would compromise the utility of the CNTs in this instance [45,46]. Covalent functionalization, however, does not include the use of these harmful reagents and even the simplest functionalization such as carboxyl groups can change the wettability of CNTs to permit dispersion in aqueous solvents [47]. Changing the physical dimensions and chemical features of CNTs can alter their inherent properties in relation to electrical conductivity and wettability, thereby improving biocompatibility [23,47]. Electrical conductivity is the property of CNTs that caters to CMs the most. Being electrogenic, these can greatly benefit cells from electrical stimulation to facilitate contraction synchronisation, cell alignment and cell maturation—something that is made possible by CNTs and derived conductive composites when they are used as tissue scaffolds [48,49,50,51,52]. As a surface coating, CNTs are able to create nano- and microscale surface roughness, which in turn acts to promote cell adhesion [53]. CNTs can also be easily combined with existing biomaterials such as hydrogels, soft plastic polymers, and matrix proteins, making it easy to phase into current research projects to adapt scaffolds for electro-active cell types.

The theoretical affinity of CNTs with cardiac myocytes makes this application intriguing for many researchers and, as a result, the amount of literature on the subject is expanding rapidly. Nonetheless, the ideal use of CNTs and a uniform methodology for constructing tissue scaffolds is not available. The problem this creates is that individual studies cannot be directly compared due to multiple experimental differences in scaffold material, design, and fabrication. It appears there is no one perfect approach in utilising CNTs for cardiac tissue engineering and several variables must be considered and factored in. Our review, therefore, aims to characterize these differences and summarise the key factors when considering the application of CNT scaffolds for cardiac myocytes. First, we aim to establish whether the use of CNT-based biomaterials are effective tools for culturing cardiac tissue ex vivo. Second, how CNT-based biomaterials can be used to exploit the unique characteristics of cardiac myocytes to bring about capabilities not offered by other tissue scaffolds and techniques. Third, where the utilization of CNTs in cardiac tissue engineering is headed in the future, based on what techniques have been most successful thus far, and finally, in what areas of cardiovascular medicine CNT-cardiac tissue engineering is predicted to have its greatest effects.

## 2. Methodology

### 2.1. Objectives

This systematic review aims to characterise the utility of CNTs as a biomaterial resource for current cardiac tissue engineering research and identify the areas in which CNTs show the most potential to advance the field.

### 2.2. Methods

The research question summarising this review was developed using the PICOT format [54] and is as follows—How has the application of carbon nanotubes in biomaterials affected cardiomyocyte culture outcomes in tissue engineering over the past 10 years? The Preferred Reporting Items for Systematic Reviews and Meta-Analysis (PRISMA) guidelines were then followed to conduct the systematic review and narrative synthesis [55].

### 2.3. Search Strategy

In October 2020, PubMed, Web of Science, and Scopus were used to search for publications using the search terms “carbon nanotube”, “cardiomyocyte”, and “cardiac myocyte”. Search terms were developed to isolate relevant publications and terms were combined into phrases in order to focus searches further. Two phrases were developed—”*cardiac myocyte **” *AND* “*carbon nanotube **” and “*cardiomyocyte **” *AND* “*carbon nanotube **”—with the stenography of the phrases being maintained between databases. Where possible, the searches were applied to all fields and databases and were refined to the last ten years from October 2020, ensuring as many areas in each database were searched as possible and the publications were as current as possible.

### 2.4. Search Attrition Criteria

The results of each phrase from each database were first sorted based on exclusion criteria. The criterion were applied as follows: The paper must be a primary research publication—no reviews, editorials, books, patents, or conference reports; the paper, or a version of, must be published in English; and the paper must be open access.

The resulting papers were then screened against the inclusion criteria that focussed on the topic of the paper based on the title and abstract. The criteria centres around the research question—how has the application of CNTs in biomaterials positively affected CM culture outcomes in tissue engineering over the past 10 years? This was distilled into the following criteria: must include the use of CNTs; must include the use of CNTs in or on tissue culture scaffolds; and must include the use of CMs—no other cardiac cell types or stem cells.

### 2.5. Article Processing and Selection

Once the exclusion and inclusion criteria had been applied to all the search results and duplicates were removed, the final publication library was reviewed by two investigators. The full texts were obtained from the respective publishers. If the full publications were not available through open access, the authors were contacted via the paper’s correspondence email. Failing that, the paper was excluded from the final review set.

### 2.6. Quality Appraisal

As laid out by the Task Force of Academic Medicine and GEA-RIME committee, the Checklist of Review Criteria was used on the final review library to assess the quality and relevance of the papers as a whole as opposed to assessing the title and abstract alone [56]. By using this framework, each section of the publications can be assessed for scientific merit as well as features that should run through the entirety of the paper such as well identified research problems, robust experimental design, and critical data analysis. The Checklist of Review Criteria categories are as follows and numbers correspond to the columns in Table 1:Problem Statement, Conceptual Framework, and Research QuestionReference to the Literature and DocumentationRelevanceResearch DesignInstrumentation, Data Collection and Quality ControlPopulation and SampleData Analysis and StatisticsReporting of Statistical AnalysesPresentation of ResultsDiscussion and Conclusions: InterpretationTitle, Authors, and AbstractPresentation and DocumentationScientific Conduct

### 2.7. Data Extraction

For this review, a specialised data extraction table was designed to collate and summarise the key information from each paper. The main categories in the table included the biological materials used (type of cells used and their source), synthetic materials used (type of CNTs, chemical functionalisation, additional scaffold biomaterials, and scaffold fabrication method), techniques used to assess CM viability and function, key results and discussion, limitations identified by authors, and future work suggested by authors.

## 3. Results

### 3.1. Search Breakdown

Search results and the following exclusion and inclusion of papers are summarised in Figure 1. In short, search phrases yielded 72 titles from PubMed, 111 titles from Web of Science, and 66 titles from Scopus (Figure 2). Of the 249 total papers, 53% were duplicates and the subsequent attrition methodology left 30 papers that focused on answering the review question.

### 3.2. Quality Appraisal

The set of 30 papers was then assessed by quality appraisal, using a total of 13 criteria. Papers that did not pass at least 12 of the 13 criteria were removed. This excluded three papers, leaving 27 out of 30 papers that were included in the review. Table 1 summarises the results of the quality appraisal and Table 2 displays the summarised contents of all 27 final publications.

### 3.3. Experimental Design

A myriad of different methodologies and techniques were used by researchers, first to fabricate CNT tissue scaffolds and second to assess the health and function of CMs cultured on the scaffolds, which are summarised for all reviewed publications in Table 2 and Figure 3.

### 3.4. Experimental Materials

The review question dictates the use of CMs, but these cells can be sourced from many different species and by numerous isolation techniques. Seventy four percent of researchers used CMs isolated from neonatal rat hearts, with 60% of those specifying isolation solely from the ventricles. CNTs also come in a variety of forms, where features can be optimised to best perform the task required. For example, SWNTs can be used to produce scaffolds with more reliable conductive properties or CNTs can be modified to possess additional functional groups that can improve hydrophilicity and even facilitate chemical cross-linking with co-polymers [87]. Of the 27 final publications, 20 used MWNTs, 22 opted to purchase CNTs from a third party, and 19 provided details on the chemical functionalisation of the CNTs used—with carboxyl functionalisation being most common among those at 47%. Seventy four percent of researchers took to combining CNTs into composites with other biomaterials in order to create scaffolds and of these additional biomaterials over half—65%—were some form of hydrogel.

### 3.5. Experimental Platforms

The design and fabrication of scaffolds is of significant importance when attempting to culture cells into functional tissues. Careful consideration and optimisation can help maximise the utility of the applied biomaterials and the cell culture outcomes. A majority of papers—74%—incorporated CNTs into biomaterials to create solid, soft scaffolds with an isotropic distribution of CNTs. Additionally, engineering two (2D) and three dimensional (3D) patterns and structures to mimic a cell’s native environment can greatly affect the ability of a scaffold to support these tissues. Some researchers have attempted to achieve this by creating pristine ‘super-aligned’ CNT sheets directly from synthesis, which aimed to direct CM growth in the direction of alignment. Meanwhile, other researchers have chosen to make composite fibres, produced by electrospinning, to make scaffolds mimicking collagen fibres observed in cardiac extra-cellular matrix. Taking advantage of CNTs’ conductivity, 30% of papers also used their scaffolds to electrically stimulate CMs. Three out of the twenty-seven publications took to combining the unique properties of CMs and CNTs to create bioinspired soft robots, capable of simple actuations [57,58,78].

### 3.6. Experimental Techniques

Immunofluorescence was most frequently used with 21 out of 27 papers employing this technique to observe specific protein expression and localization. All of these papers chose to stain for alpha actinin as a cardiac specific marker as well as to observe the organisation of contractile protein complexes. In addition, 76% chose to stain for connexin-43 and 48% also stained for cardiac troponin I, with 24% also employing western blot to back up and quantify observations seen by immunofluorescence. Other commonly used techniques included morphological and histological analysis, visual and mechanical contractility analysis, calcium imaging, electrophysiological studies, and cell viability and metabolism assays (Figure 3).

### 3.7. Experimental Results

All papers—irrespective of the materials, fabrication methods, and cellular assessment techniques used—showed that CMs grew preferentially on scaffolds containing CNTs when compared to the controls. Twenty six percent of papers, through nano- and microscale alignment of CNTs and accompanying scaffold material, also showed controlled growth and elongation of CMs in the direction of the scaffold alignment. It is also worth mentioning that while some scaffolds incorporating CNTs had a positive effect on CM culture when compared to the experimental controls, higher concentrations of CNTs can cause dose-dependent cytotoxicity, which was reported by six out of 27 papers. Seventy four percent of papers reported increases in the expression of proteins relating to cardiac function such as alpha actinin, troponin I, and connexin-43. In addition, 70% of researchers stated CMs exhibited improved organisation and alignment of structural, sarcomeric proteins with 26% specifically referring to the improved localisation of connexin-43 to electrical junctions between cells known as intercalated discs. As seen by numerous different techniques mentioned briefly in the previous section, CMs also exhibited enhanced contractile and electrical maturity on CNT scaffolds based on cell elongation, formation, and organisation of sarcomeres and intercalated discs, beating rate, beat synchronisation of cell sheets, action potential duration, and resting membrane potential, among other parameters, were all improved.

## 4. Discussion

Tissue engineering has provided vast opportunities to develop more robust platforms for cardiac research. Central to this is the continued need to improve experimental design and execution through advances in materials science technology.

Part of the rationale behind the use of CNTs in cardiac tissue scaffolds is their resemblance to the ECM protein type I collagen; a single triple helix typically being ~2 nm in dimeter and ~300 nm in length [88]. In the cardiac microenvironment *in vivo*, the ECM is not simply a passive and inert scaffold but is in fact crucial in the governance of cell function, metabolism, and survival by biophysical and biochemical cues, which is something that has been drastically overlooked until recently [89,90]. Standard protocols for CM cell culture normally utilise either glass or plastic surfaces coated with proteins such as laminin, fibronectin, and poly-d-lysine [17,91]. While these proteins may be present in the cardiac ECM, these culture substrates are missing the dynamic and complex network of multiple ordered proteins, which forms the basis of CMs’ mechanosensitive adherence and ability to form organised electrical and contractile protein complexes [89,90]. The technology to perfectly replicate the contents and intricacy of cardiac ECM artificially is currently lacking, but CNTs, alone or combined with additional materials, have demonstrated their capacity to support CMs’ structure and function as a result of their unique mechanical, chemical, and electrical properties [92].

One of the most interesting, and recent, developments seen in the literature was the use of CNTs and their composites in the creation of bioinspired soft robots. A limited number of our publication cohort directed research efforts toward this application, which in this case adds to the novelty of their findings. In 2015, Shin et al. aimed to design a “3-D bio-hybrid actuator” where precise and reliable control over the contractile elements of the tissue construct could be established by electrical stimulation in vitro [78]. Even distribution of biomaterial conductivity was recognised as a key parameter and as such, aligned CNT forest electrodes were imbedded at regular intervals in the scaffolds as well as a surface CNT-GelMA composite layer onto which CMs would be cultured. Integration of electrical stimulation protocols into cell culture enabled control over CM contractions and lowered CM excitation thresholds when stimulation was applied parallel to the alignment of the CNT electrodes. Significant movement, as in the distortion of the scaffold’s shape or movement of the actuator from one place to another, was limited in this research, however, this laid a foundation for work published by the same research group in 2020. In Wang et al. (2020), design of the soft robot was based on the shape of the manta ray [58]. The triangular, winged structure was identified as a simple and achievable design capable of repetitive contractions that could allow for directional movement in vitro. With even conductivity ensured by flexible gold electrodes imbedded in the scaffold, the key parameter identified in this research was the spacing of the ridged micropattern on the CNT-GelMA layer, which hugely influenced the direction and synchronicity of CM contractions. Spacing at 75 µm caused CM alignment in the direction of the ridges but maintained an interconnected monolayer, resulting in the wings of the soft robot contracting in unison and achieving intentional movement through the culture medium even without electrical stimulation. The addition of electrical stimulation, either through the culture medium or directly through the gold electrodes, allows for moderate control over the soft robot’s actuations. The last publication on cardiac soft robots took inspiration for its design from the caterpillar. Unlike Wang et al. (2020), Sun et al. (2020) designed their soft robot to be grounded, rather than swimming, with the caterpillar design moving along a track aided by snake-like asymmetric claws [57]. The design included a magnetised GelMA layer coated on top with an iridescent colour layer, so movement could be more easily observed, and on the clawed bottom with aligned CNTs, as a conductive substrate which could also guide cell alignment. Research application aims were more clear than previous publications as isoproterenol stimulation and hyperkalemia were employed to demonstrate how the tissue construct behaved as a representative disease model for cardiac muscle where contraction rate and progress along the PDMS ‘racetrack’ could be used as metrics for the efficacy or toxicity of drugs.

### 4.1. The Effects of Carbon Nanotubes on Cell Structure

In any cell type, structure is intrinsically linked to function. Making simple observations assessing morphology against a well-characterised standard is an effective, albeit very limited, indicator of cell health. However, this assessment is subjective and reliant on sufficient professional expertise, but observations can be reinforced and quantified by computational solutions such as ImageJ [93] and fast Fourier transform analysis as employed by some papers [59,62,63,64,67,70,73,77,78,80,85]. CMs in the native myocardium exist as continuous, unidirectional sheets connected by end-to-end intercellular junctions, which optimises whole tissue contraction [70]. Under in vitro conditions, cells will still attempt to form these sheets and contract synchronously [94]. In the case of CMs, observing these behaviours can also be an indicator of cellular development or maturity as well as cell health. Several papers reported, either observed or measured by multiple techniques, enhanced cell elongation and alignment on CNTs scaffolds against controls—irrespective of the materials used—simply indicating the addition of CNTs to any number of substrates or materials provides physical and/or chemical stimuli for standard CM morphology. In papers that employed microscale composite fibre and nanoscale CNT aligned architecture in their scaffolds, [57,64,70,80] observed further improved myocyte alignment against control scaffolds with random, unaligned architecture. It is also important to recognise that CM morphology changes significantly as cells mature, which is of particular interest in this review as a large majority of papers used either neonatal or human induced pluripotent stem cell-derived CMs, where both present smaller, mononucleated, and more circular morphologies indicating incomplete maturation [95,96]. Thus, compounding the already observed effects of CNTs on cardiac tissue scaffolds with their ability to elicit a more mature phenotype from immature cell types.

Assessment of CM morphology and alignment was frequently combined with immunofluorescence to resolve the assembly and arrangement of intracellular proteins, providing greater detail on cellular responses to their environment. Contractile proteins commonly stained for included alpha actinin, and cardiac troponin T and I; these proteins form complexes as part of the contractile elements in CMs known as myofibrils. Myofibrils are further divided into repeating units known as sarcomeres, the borders of which are defined by features called Z bands. Alpha actinin facilitates the attachment of actin filaments to the Z bands and can be used to observe the striation of myofibrils as a sign of mature CM protein organisation [97]. Furthermore, the cardiac troponin complex, consisting of subunits C, I, and T, are positioned along the length of the actin filaments in myofibrils, which block the binding site for myosin during diastole. Staining of these proteins enables visualisation of the longitudinal arrangement of myofibrils, further indicating the level of CM alignment and maturity [98]. On top of the effects seen on external structure and morphology, CNTs were also observed to improve the internal organisation and alignment of these mechanical and contractile proteins. Similar to external morphological investigation, staining of these proteins was quantified by computational tools like ImageJ [59,62,63,64,73,77,80,93].

CMs have been shown to have a dynamic relationship with their environment via their mechanosensory focal adhesions [99,100]. The features of the extracellular environment can affect cell motility, protein expression, and even phenotype, but also, most importantly in this case, the arrangement of structural proteins. With the geometrical similarity between CNTs and collagen fibres, the main component of native cardiac ECM, scaffolds including CNTs may provide the necessary stimulus for strong, sensory CM adhesion. Few papers have assessed the adhesion of CMs on scaffolds and those that did either performed rudimentary visual observation of cells [65,82] or employed phalloidin to stain for F-actin; a structural protein that indicates the areas of the cell membrane that have adhered to the substrate [74,79,85]. While this may highlight the structures of the cytoskeleton giving an impression of where focal adhesions have formed, this is not a direct assessment and proteins actually contained in focal adhesion complexes could be stained and imaged instead.

The aptitude of CNTs to modulate CM structure is clear, but how does this relate to tissue engineering and ultimately the restoration of research into cardiovascular diseases? The aim of experiments into cardiac tissue scaffolds has largely been to mimic what is observed naturally in cardiac muscle and for the most part, this has been achieved in multiple aspects of CM morphology. However, the planned application of many of these technologies is as treatment for infarct, damaged heart tissue, which has limitations of its own and compounding this with the toxicity of CNTs in whole organisms makes research of this ilk ultimately redundant. The frequently over-looked exploit of this revelation is that CNTs have afforded researchers a degree of control over CMs. While a healthy, functional state in CMs has so far been the aim, so too could a diseased, dysfunctional state be attained and perhaps be of greater use to researchers. CNTs, in this way, could be better put to use in generating disease states in CMs, which would provide a platform for assessing biologically representative pathologies pharmacologically or otherwise—also reducing the need for animal models and sick patients. Though simply a postulation, the principle remains the same that through appropriate optimisation and application of CNT scaffolds can be used to generate healthy and diseased CMs that would revolutionise cardiac research. For this aspiration to be successful, it is also essential that we understand how exactly CNTs affect CM structure, whether this is through mechanical, chemical, or electrical stimuli.

### 4.2. The Effects of Carbon Nanotubes on Cell Function

With their conductive properties and likeness to native cardiac ECM, CNTs have been seen to improve CMs’ contractile and electrical function across the board, manifested in several ways, and observed by multiple techniques and measurements. Having only recently been introduced to tissue engineering, generally and within the spheres of cardiac research, the exact relationships between cellular functions and CNTs is still largely unclear. However, publications retrieved in this review have made strides in broadening this understanding.

Publications by Sun et al. between 2015 and 2017 interrogated the metabolic mechanism behind the formation of structures known as intercalated discs [73,79], which facilitate the electrical and mechanical intercellular connections between CMs [101]. Pristine SWNTs were shown to increase the expression of connexin-43 and N-cadherin—both proteins found in intercalated disc structures—and increase the assembly of intercalated discs. This was found to be regulated by focal adhesion and extracellular signal-regulated kinase pathways, FAK and ERK, respectively, with both also being regulated by β1-integrin [79]. In subsequent experiments, β1-integrin was also shown to activate RhoA, a small GTPase shown to be involved in regulating cell-cell junctions [73,102]. Furthermore, it was elucidated that the FAK pathway regulated the expression of connexin-43 and the RhoA pathway regulated the expression of N-cadherin [73]. As connexin-43 is an integral protein regarding the electrical function of CMs, it can be said that FAK regulates, to an unknown extent, the electrical function of CMs on CNTs [103]. Similarly, through affecting the expression of N-cadherin, RhoA can be said to effect the mechanical function of CMs [104]. What is most encouraging about this for cardiac tissue engineering is that β1-integrin functions in a similar fashion in the native myocardium, thus giving credence to the theorised likeness of CNTs to cardiac ECM [105,106].

CMs are particularly difficult to culture and manipulate outside of the body as they are terminally differentiated, meaning that they lose the ability to proliferate once fully mature, which leaves the heart unequipped to repair itself after trauma [107]. Additionally, after traumatic events, the damaged area can scar, making it stiff and avascular, which induces local CMs to dedifferentiate into fibroblasts—a cell type with no contractile function and no natural way of returning to the CM phenotype [108,109]. Many papers set this precedent, so specifically investigated the capability of CNTs to first maintain the CM phenotype of cells based on protein expression, and second in bringing about a more proliferative state. Researchers frequently used α-actinin, an important protein that forms part of the contractile machinery of CMs, to distinguish CMs from other cell types in culture and on occasion, this was also supplemented with other cardiac-specific proteins like cardiac troponins I and T. One research group was able to culture multiple cardiac cell types on their scaffolds as demonstrated by positive staining for fibroblasts and endothelial cells in addition to CMs [69]. A combinative assessment from all techniques used provided researchers with convincing evidence that CNTs and scaffolds containing CNTs create an environment for CMs, which fosters maintenance of their electrically and mechanically active phenotype. Fewer papers have incorporated proliferative assessment, however, those that did also found, either through the use of fluorescently-labelled nucleotides or quantification of DNA, that CMs were multiplying to a greater extent on CNT scaffolds [69,79,85,86]. This discovery represents a huge milestone in cardiac tissue engineering and regenerative medicine, and while CNT scaffolds may not be appropriate for implantation, they have now created a platform for the assessment of CM phenotype and proliferation stimuli that could pave the way for novel therapeutic treatments.

While CNTs show great promise in cardiac tissue engineering, it is vitally important to recognise their pitfalls as well as their successes when imagining how CNTs will be best applied in biosciences and medicine. It is frequently predicted by these and many other publications that cardiac tissue scaffolds hold the most potential in regenerative medicine, either through the delivery of new cells to damaged, infarcted areas of the heart or in supplementing cardiac function as an additional synthetic layer of myocardium for mechanical support and pace making. Many researchers will agree that this is the optimal treatment post-myocardial infarction, though must also concede that CNTs possess a fatal flaw that does not permit their therapeutic use *in vivo*—cytotoxicity. Before and during the scope of this review, many publications revealed that CNTs had both physical and chemical mechanisms of toxicity. Poland et al. (2008) demonstrated that CNTs possess “asbestos-like” toxicity by penetrating cell membranes, which also contributed to genotoxicity as nanosized structures can enter the nucleus [110,111]. CNTs then enact chemical damage through the creation of reactive oxygen species, which induces oxidative stress in cells—namely in the nucleus and mitochondria [112]. It was also later found that CNT cytotoxicity is dose-dependent in the respect that cell death shifts from apoptosis at low CNT concentrations to necrosis at high concentrations [113]. Some publications found by this review commented on the dose-dependent nature of CNT toxicity in their experiments, but rather concerningly, this was not enough to deter the aim of using scaffolds in patients in many cases. Though a large amount of information on CNT cytotoxicity has been published before in many of the reviewed papers, researchers still insisted on the use of their scaffolds *in vivo*, yet by assessing papers chronologically, a change in professional opinion could be observed after 2017 where these suggestions are superseded by calls for scaffolds to be used in in vitro models for cardiac pathologies and drug development. One such example is Wang et al. (2018), where researchers designed their scaffolds with the express intent of creating a platform for drug development [67].

An overriding concern when taking a broader look at the publications retrieved in this systematic review is the proportionate absence of electrophysiological studies into CMs to interrogate cell function. Many techniques were employed by researchers to assess CM function but, being bioelectrogenic entities, it is ultimately best defined through characterising ion channel functionality. Electrophysiological studies using microelectrode approaches provide insight into the true phenotypic properties at the single cell and tissue level [114]. Only six out of 27 papers employed whole cell patch clamping studies in their research with an additional publication using their scaffolds to record cell electrophysiology [62,65,70,72,84,86]. Meanwhile, pseudo-electrophysiological techniques were commonly used by researchers—largely fluorescent imaging of calcium transients with some also using voltage-sensitive fluorescent dyes like Di-8-ANEPPS—indicating perhaps that researchers acknowledge the importance of this kind of data but lack the equipment or expertise to collect it [63,66,69,71,73,74,79,81,82,84]. Alternatively, it could be that current tissue scaffolds are not able to facilitate patch clamping studies either due to the incompatibility of scaffolds with equipment or the potential inability to accommodate a sufficient number and calibre of CMs. Regardless, future work in cardiac tissue scaffolds must begin to integrate true electrophysiology studies into research as without this, cells may only appear to behave as native CMs while potentially possessing atypical intrinsic function.

## 5. Limitations of This Study

It is important to acknowledge the limitations of this review to properly understand and appreciate the conclusions drawn and how they relate to the wider fields of nanotechnology and tissue engineering. Some of these limitations are intrinsically linked to the review methodology and to general attitudes toward publishing research. In this review, only three publication databases—PubMed, Web of Science and Scopus—were searched, also, only English and open access publications were considered. Therefore, the 27 papers retrieved through our methodology may not form a representative summary of the current information and research on CNT-based scaffolds for CM tissue engineering. Additionally, it is well known that studies with positive results, which successfully prove experimental hypotheses, are more heavily represented in the literature than those studies with negative results—forming what is commonly known as “publication bias” [115]. Therefore, in the context of this review, the number of instances in which CM cell culture has been positively influenced by the addition of CNTs may be artificially inflated.

Moreover, the fields of CNT biomaterials and CM tissue engineering are still considered to be pioneering and, in their infancy, when compared to other more established areas of research. Without a well-defined foundation among researchers, the techniques and methods of assessing, first, the physical properties of the CNT scaffolds and, second, the biological state of cells grown on those scaffolds are diverse and inconsistent, meaning that directly drawing comparisons between results from different publications would be disingenuous and misleading. Having said this, it can be seen from the results of this review that discrete publications and research groups are beginning to use similar techniques, nonetheless, indicating that as this type of research matures, so too will the consistency of the methodology of publications.

## 6. Conclusions

CNTs are an incredibly versatile material and have shown promise in many fields in recent years from cellular imaging in micro-CT devices to vehicles for targeted drug delivery [116]. From a seemingly unlikely mix, CNTs have shown merit, particularly in tissue engineering, being used in a wide variety of cell types and for a breadth of applications; for one, allowing the creation of smart, tunable scaffolds for cell culture. Their application has arguably been of most significance in the engineering of electrogenic cell types, specifically cardiac myocytes. The properties of CNTs appear to naturally pair with the needs of CMs and their practical combination has certainly proven this hypothesis to be true. CNT tissue scaffolds have, like never before, provided CMs with stimuli that recapitulate the endogenous states seen *in vivo*—mechanically, electrically, and genetically. Limited toxic effects were observed for all papers retrieved in this review and the revolutionary effects brought about solely by the CNTs warrant continued effort in seeking their optimal use. As a material that effectively simulates the natural environment of the heart and can modulate cell behaviour, CNT scaffolds could be successfully applied in reliable models for cardiac pathologies and novel pharmaceuticals, thus reducing dependency on animal models and clinical trials. In the UK alone in 2018, approximately 74,000 animals were used in research and pre-clinical trials for CVD with roughly 13,000 of those being sacrificed for research [117]. Through these models, a greater understanding of the cardiovascular system could be gleaned, which could perhaps provide therapies that negate the need for surgical and transplantative therapies for CVD and reinvigorate this struggling but enormously important field.

## Figures and Tables

**Figure 1 bioengineering-08-00080-f001:**
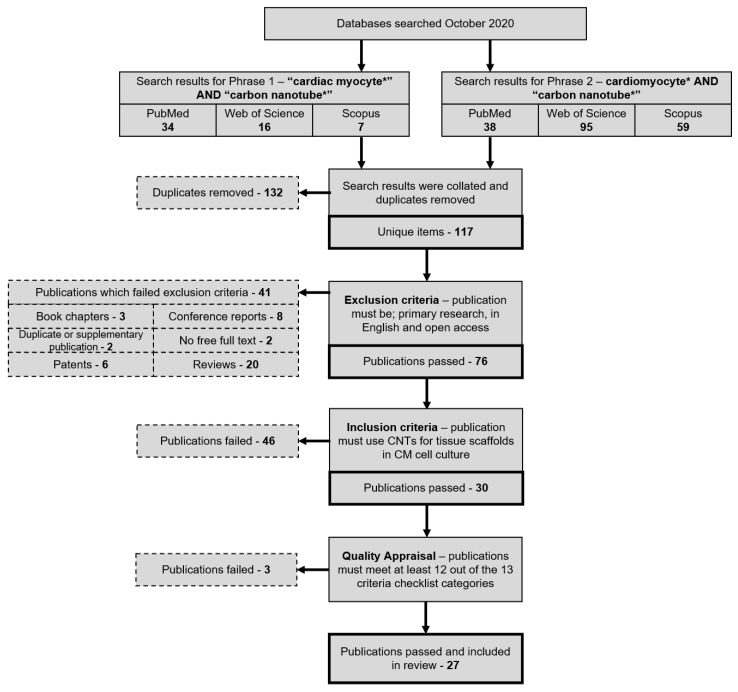
Publication attrition process depicted in a step-by-step flowchart that details the method by which the final set of reviewed publications was produced. PubMed, Web of Science, and Scopus publication were searched using one of two search phrases to gather publications relating to CNTs and CMs. After removing duplicate items, exclusion, inclusion, and quality appraisal criteria were applied to isolate primary research publications specifically relating to the review question.

**Figure 2 bioengineering-08-00080-f002:**
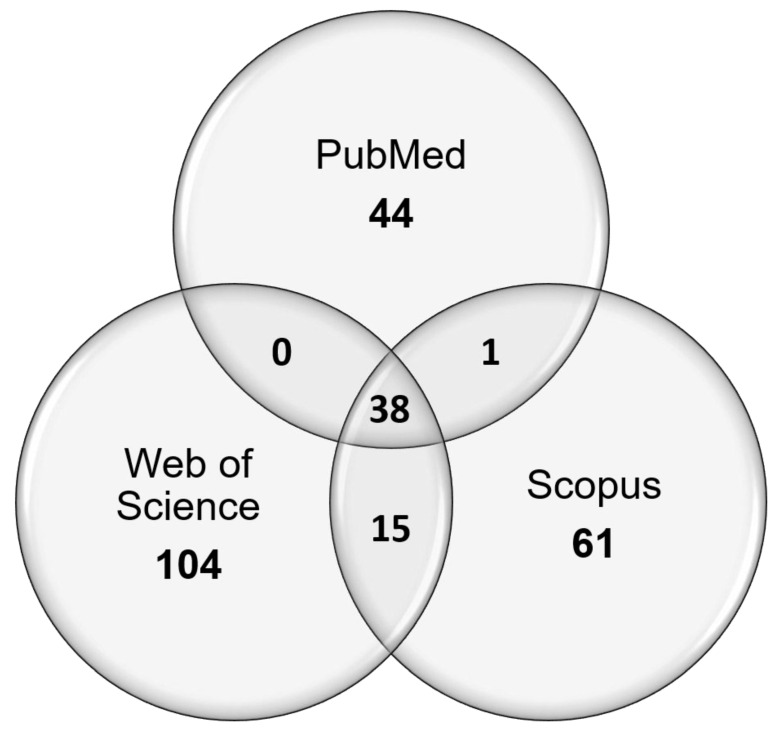
Venn diagram depicting the sources of unique publications from searches using search phrases 1 and 2, and the overlap between the different chosen publication databases.

**Figure 3 bioengineering-08-00080-f003:**
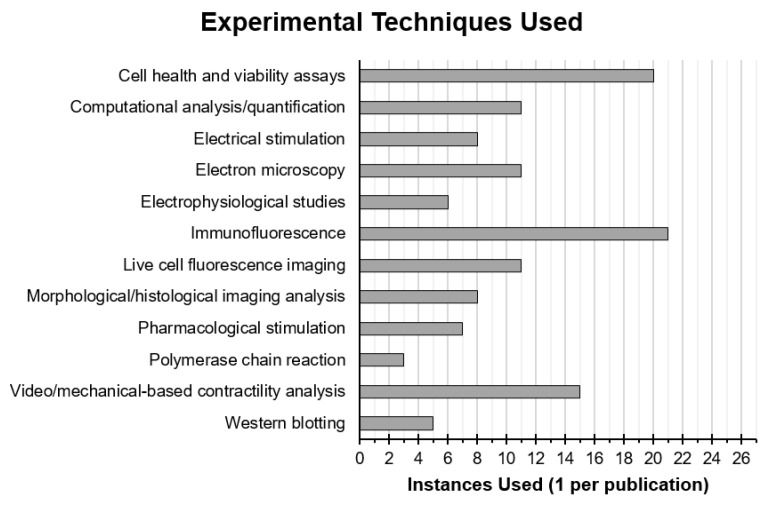
Instances of experimental techniques used in reviewed publications that were employed either to interrogate the health and function of CMs or their interaction with CNT-based scaffolds. Techniques using similar principles and materials or that produced similar data were put into groups.

**Table 1 bioengineering-08-00080-t001:** Quality appraisal of the 30 publications retrieved after database inception and search attrition. (1) Problem Statement, Conceptual Framework, and Research Question, (2) Reference to Literature and Documentation, (3) Relevance, (4) Research Question, (5) Instrumentation, Data Collection, and Quality Control, (6) Population and Sample, (7) Data Analysis and Statistics, (8) Reporting of Statistical Analyses, (9) Presentation of Results, (10) Discussion and Conclusion: Interpretation, (11) Title, Authors and Abstract, (12) Presentation and Documentation, (13) Scientific Conduct.

Publication	Categories of Checklist of Review Criteria [56]													Total Criteria Met
	1	2	3	4	5	6	7	8	9	10	11	12	13	
Sun 2020 [57]	✓	✓	✓	✓		✓	✓	✓	✓	✓	✓	✓	✓	12
Wang 2020 [58]	✓		✓	✓	✓	✓	✓	✓	✓	✓	✓	✓	✓	12
Zhao 2020 [59]	✓	✓	✓	✓	✓	✓	✓	✓	✓		✓	✓	✓	12
Alvarez-Primo 2020 [60]	✓	✓	✓	✓	✓	✓	✓	✓	✓		✓	✓	✓	12
Chen 2019 [61]	✓	✓	✓	✓		✓	✓	✓	✓		✓	✓	✓	11
Lee 2019 [62]	✓	✓	✓	✓	✓	✓	✓	✓	✓		✓	✓	✓	12
Roshanbinfar 2019 [63]	✓	✓	✓	✓	✓	✓	✓	✓	✓		✓	✓	✓	12
Vaithilingam 2019 [64]	✓	✓	✓	✓	✓	✓	✓	✓	✓		✓	✓	✓	12
Hou 2018 [65]	✓	✓	✓	✓	✓	✓	✓	✓	✓		✓	✓	✓	12
Martinelli 2018 [66]	✓	✓	✓	✓	✓	✓	✓	✓	✓		✓	✓	✓	12
Wang 2018 [67]	✓	✓	✓	✓	✓	✓	✓	✓	✓		✓	✓	✓	12
Ahadian 2017 [68]	✓	✓	✓	✓	✓	✓	✓	✓	✓	✓	✓	✓	✓	13
Peña 2017 [69]	✓	✓	✓	✓	✓	✓	✓	✓	✓		✓	✓	✓	12
Ren 2017 [70]	✓	✓	✓	✓	✓	✓	✓	✓	✓	✓	✓	✓	✓	13
Roshanbinfar 2017 [71]	✓	✓	✓	✓	✓	✓	✓	✓	✓		✓	✓	✓	12
Sherrell 2017 [72]	✓	✓	✓	✓	✓	✓	✓	✓	✓		✓	✓	✓	12
Sun 2017 (1) [73]	✓	✓	✓	✓	✓	✓	✓	✓	✓		✓	✓	✓	12
Sun 2017 (2) [74]	✓	✓	✓	✓	✓	✓	✓	✓	✓		✓	✓	✓	12
Wu 2017 [75]	✓	✓	✓	✓		✓	✓	✓	✓		✓	✓	✓	11
Yu 2017 [76]	✓	✓	✓	✓	✓	✓	✓	✓	✓		✓	✓	✓	12
Liu 2016 [77]	✓	✓	✓	✓	✓	✓	✓	✓	✓		✓	✓	✓	12
Shin 2015 [78]	✓	✓	✓	✓	✓	✓	✓	✓	✓		✓	✓	✓	12
Sun 2015 [79]	✓	✓	✓	✓	✓	✓	✓	✓	✓		✓	✓	✓	12
Kharaziha 2014 [80]	✓	✓	✓	✓	✓	✓	✓	✓	✓		✓	✓	✓	12
Pok 2014 [81]	✓	✓	✓	✓	✓	✓	✓	✓	✓	✓	✓	✓	✓	13
Tao 2014 [82]	✓	✓	✓	✓	✓	✓	✓	✓	✓	✓	✓	✓	✓	13
Zhou 2014 [83]	✓	✓	✓	✓		✓	✓	✓	✓		✓	✓	✓	11
Martinelli 2013 [84]	✓	✓	✓	✓	✓	✓	✓	✓	✓		✓	✓	✓	12
Shin 2013 [85]	✓	✓	✓	✓	✓	✓	✓	✓	✓		✓	✓	✓	12
Martinelli 2012 [86]	✓	✓	✓	✓	✓	✓	✓	✓	✓	✓	✓	✓	✓	13
											**Total Passed**			**27**

**Table 2 bioengineering-08-00080-t002:** Summary of the contents of all 27 reviewed publications—author’s abbreviations: CM (cardiomyocyte), ePhys (electrophysiology), eStim (electrical stimulation), IF (immunofluorescence), pStim (pharmacological stimulation), WB (western blotting).

Publication	Materials Used				Techniques Used to Assess CM Viability & Function	Results & Discussion	Limitations Suggested by the Authors	Further Work Suggested by the Authors
	Biological		Synthetic					
	Cells	Source	CNTs	Scaffold Material and Design				
Sun 2020 [57]	Neonatal cardiomyocytes	1- to 2-day old rat pups	No information given	Caterpillar-inspired soft robot consisting of a magnetised GelMA layer with asymmetric claws, an aligned CNT-coated GelMA layer and an iridescent colour layer.	Confocal imaging of F-actin plus DAPI, MTT assay, SEM, visual analysis of CM contractions and soft robot actuation, pStim by isoproterenol, disease modelling by induced hyperkalemia.	Aligned CNTs improved electrical and mechanical properties of the scaffold, as well as improving biocompatibility by increasing cytoskeletal alignment and cell elongation. CM beating frequency was adversely affected by increased scaffold thickness and increased stiffness, influenced by GelMA percentage. pStim increased CM beating and speed of soft robot movement along PDMS “racetrack”. Increasing potassium concentration to induce hyperkalemia caused CM contractions to slow down and eventually stop, however pStim caused continuation of beating.	Only hyperkalemia tested, while there are many other “varieties of heart disease in actual clinical practice”. The microfluidics system was expected to perform more complex functions.	Further research needs to improve the platform and explore more applications for soft robots in clinical treatments.
Wang 2020 [58]	Neonatal cardiomyocytes	2-day old Sprague Dawley rat pups	Purchased MWNTs—carboxylic functionalisation	Manta ray-inspired soft robot consisting of a gold electrode embedded between micropatterned PEGDA and GelMA-MWNT hydrogel layers.	Confocal IF of α-actinin and Cx43, video recording of CM contractions and soft robot actuation, field and direct eStim.	Precise spacing of the CNT-GelMa micropattern improved CM orientation and alignment—75µm spacing induced alignment and CMs were interconnected for synchronous beating. CMs on CNT-GelMA showed good alignment and elongation as well as uniaxial sarcomere alignment with interconnecting structures between cells. This was improved where cells were directly above the gold electrodes. eStim allowed a moderate amount of control over soft robot actuations without harming CMs. CNT cytotoxicity was acknowledged but said to be limited by only using CNT concentrations up to 5 mg/mL in the scaffold.	Parts of the production process are technically very challenging.	Optimise the fabrication process to improve ease of production.
Zhao 2020 [59]	Neonatal ventricular cardiomyocytes	Neonatal rat pups	Purchased MWNTs—98% purity, 10–30 µm length, 10–20 nm diameter, carboxylic functionalisation.	Random and aligned electrospun fibres consisting of *Bombyx mori* silk fibroin, PEO, gum arabic and varying concentrations of CNTs.	Live/Dead assay, confocal IF of α-actinin, cTnI, Cx43 plus F-actin and DAPI, custom software analysis of CM alignment, ImageJ analysis of nuclear aspect ratio and IF staining.	Cell viability maintained at >90% at day 7 showing insignificant effect from CNTs, solvent-free method and the nanofibrous morphology of the scaffolds. Aligned scaffolds guided cell growth parallel to the fibre direction. Scaffolds containing CNTs increased cell elongation. Aligned scaffolds induced more rod-shaped CM morphology and anisotropic organization. Scaffold alignment was more dominant in this regard than CNT content. CNTs increased sarcomeric organization and expression of α-actinin and cTnI. Cx43 expression was increased in CNT scaffolds and with scaffold alignment.	Challenge remains of producing entirely interconnected networks of gap junctions in engineered cardiac tissues which will function normally when transplanted and not induce arrhythmia.	No further work suggested.
Alvarez-Primo 2019 [60]	AC16 human cardiomyocytes	Derived from the fusion of primary ventricular cardiomyocytes and fibroblasts	Purchased SWNTs—pristine, produced by HiPco process	SWNTs dissolved in three different surfactants—SDS, CTAB and PF108—were mixed with sodium alginate to produce hydrogel scaffolds.	Live/Dead assay	80% cell viability was observed in scaffolds which used PF108, and no cells, alive or dead, were detected on the other scaffolds.	No limitations given.	Assess whether PEO, poly(pyrrole), poly(thiopene) or poly(aniline) could be used as dispersion agents for SWNTs in tissue engineering applications. Development of biodegradable SWNT-alginate hydrogels.
Lee 2019 [62]	Neonatal cardiomyocytes	2-day old Sprague Dawley rat pups	Purchased MWNTs—95% purity, 30 ± 15 nm length, 5 ± 20 nm diameter, carboxylic functionalisation	CNTs, along with GO and rGO, were dispersed in GelMA to produce thin hydrogel films.	PicoGreen assay, confocal IF of vinculin, α-actinin, cTnI, Cx43 plus F-actin, CellProfiler, ImageJ, and MATLAB analysis of IF staining, video recording and MATLAB analysis of CM contractions, excitation threshold testing by eStim, whole cell patch clamp, pStim by blebbistatin, qRT-PCR of Cx43, α-actinin, cTnI, integrin subunits α1, α2, α3, α5, α7, and β1, melusin, dystrophin, β-actin, α-tubulin, RhoA, Rac1, PAK, CDC42, TAZ, vinculin, YAP and SRC.	CMs on CNT and rGO films displayed more classic CM morphology, with increased elongation and cell spreading. Surface chemistry of nanomaterials had a greater effect on cell morphology than their geometry. CNT scaffold maintained highest cell viability after 5 days in culture. CNT-GelMA suggested as optimal nanomaterial scaffold based on cell maturation, retention, and viability. CNT and rGO scaffolds showed increased expression of all IF stained proteins compared to GO. CMs on CNTs displayed markedly increased upstroke velocity and action potential duration. Based on action potential features, cells on CNTs more closely mimic ventricular CMs and GO cells more closely mimic atrial CMs, while rGO cells displayed intermediate characteristics. CMs on CNTs and rGO had lower excitation thresholds and could be externally paced after 1 day. CNTs showed increased expression of Cx43 and α-actinin compared to native myocardial tissue, however no difference in terms of cTnI expression. CMs on CNT scaffolds showed increased expression of integrin subunits α1, α2, α3 and β1 after 3 and 5 days of culture, while rGO CMs showed increased expression of α5, α7 and β1 after the same time periods. CMs are commonly known to use integrin heterodimers α1β1 and α2β1 to bind to cardiac ECM proteins. From qRT-PCR of cytoskeletal, Rho family, mechanotransduction and growth proteins, CMs on CNTs showed increased expression of proteins related to growth and mechanotransduction as well as melusin, RhoA and Rac1.	Fibroblast contamination in isolated cells may have contributed to DNA content in PicoGreen assay, therefore skewing results.	No further work suggested.
Roshanbinfar 2019 [63]	Human induced pluripotent stem cell (hiPSC)-derived cardiomyocytes	hiPSCs differentiated by CHIR-99021-IWR-1-*endo* protocol	Purchased MWNTs—carboxylic functionalisation modified to carbodihydrazide	Thermoresponsive pericardial tissue-derived hydrogel with CNTs dispersed at 0.5 wt% concentration.	Live/Dead assay, Kymograph and MUSCLEMOTION analysis of CM contractions, IF of cTnI, α-actinin and Cx43 with ImageJ analysis plus DAPI, calcium imaging with Fluo-4, pStim by isoproterenol and epinephrine.	CNTs showed similar viability to Matrigel. Kymography showed synchronous beating on all scaffolds but only CNTs didn’t show arrhythmia. MUSCLEMOTION analysis showed increased contraction amplitude, speed, and BPM, which was enhance but not pathologically by pStim. IF showed no change in cTnI, increase in Cx43 and increased sarcomere length from α-actinin staining. ImageJ analysis of cTnI IF showed improved unidirectional alignment on CNTs. Calcium imaging showed increased BPM, peak amplitude and faster excitation and relaxation. Long term repeat experiments showed no change in IF and pStim results.	No limitations given.	No further work suggested.
Vaithilingam 2019 [64]	hiPSC-derived cardiomyocytes	Adult male human skin biopsy	Purchased MWNTs—carboxylic functionalization	PETrA was combined a curing agent and 0.1 wt% CNTs and was 3D printed by 2PP to create aligned, ridged scaffolds.	Live/Dead assay, IF of α-actinin plus F-actin and Hoechst 33258, eStim, ImageJ analysis of IF staining.	IF showed increased alignment and length of sarcomeres. eStim was shown to negatively affect viability in a voltage-dependent manner but low voltage stimulation and 3D scaffold architecture improved myofibril organization. Alignment and aspect ratio were shown to be improved on 3D scaffolds.	Creation of nanoscale scaffold topography wasn’t possible with the ink containing CNTs.	No further work suggested.
Hou 2018 [65]	Embryonic atrial & ventricular cardiomyocytes	E11 stage White Leghorn chicken embryos	Synthesized SWNTs—produced by CVD into thin films	*Nephila clavata* silk fibres wrapped in aligned CNT sheets.	SEM, ePhys via scaffold, pStim by isoproterenol.	SEM showed tight contacts between hybrid fibres and CMs. ePhys via scaffolds revealed increased long-term health and maturity of cells based on action potential amplitude and frequency.	No limitations given.	No further work suggested.
Martinelli 2018 [66]	Neonatal ventricular cardiomyocytes	0- to 1-day old Wistar rat pups	Purchased MWNTs—organic functional groups added using sarcosine and heptanal.	3D porous PDMS scaffold, surface coated with CNTs.	Alamar Blue assay, EdU assay, confocal IF of α-actinin, cTnI and Cx43 plus Hoechst 33342, calcium imaging by Fluo-4.	Addition of CNTs to 3D-PDMS increased cell retention and viability. Based on IF, CMs in CNT scaffolds were “phenotypically different” from the control scaffold. IF of cTnI showed improved sarcomeric organisation. CNTs increased Cx43 expression and localization to gap junctions. CMs in CNT scaffolds showed increased beating rates with more rhythmic oscillations in intracellular calcium. 8% increase in CM proliferation on CNT scaffolds from controls with no change in proliferation of fibroblasts.	Possible cell ageing, or cytotoxicity, induced by MWNT-coated scaffolds.	No further work suggested.
Wang 2018 [67]	hiPSC-derived cardiomyocytes	Purchased from Cellular Dynamics	Purchased CNTs—20–30 nm diameter, 10–30 µm length (MWNTs based on diameter)	Onto a 3-well PDMS culture chamber, 1:5 CNT:PDMS strips were adhered across the culture membrane. On top of which a PDMS-fluorescent bead layer was spin coated.	Confocal IF of α-actinin and MYH7, calcium imaging by Fluo-5, pStim by isoproterenol, verapamil, omecamtiv mecarbil, ivabradine and E-4031, impedence spectroscopy and Poincaré plot analysis of CM contractions.	IF showed increased alignment which improved over time but there was no change in sarcomere length. Calcium imaging showed similar peak intensity, frequency, and duration to controls. Scaffold resistance was shown to increase over time as cells began to adhere and become electrically active. CMs were also shown to contract more regularly as variability between beats decreased. pStim showed all drugs affected CMs as expected, demonstrating the scaffolds’ effectiveness as a drug testing platform.	No limitations given.	No further work suggested.
Ahadian 2017 [68]	Neonatal cardiomyocytes	1- to 2-day old Sprague Dawley rat pups	Purchased MWNTs—40–90 nm diameter, 10–20 µm length, organic functional groups added using nitric and sulphuric acid	PEGDM-124 polymer mesh scaffolds with CNT concentrations of 0.1, 0.5 and 1.5%, coated with 2% gelatine solution.	Live/Dead assay, confocal IF of α-actinin plus F-actin, eStim.	0.1 and 0.5% CNT scaffolds began beating 2 days earlier than controls, and more rhythmically and synchronously. No difference seen from Live/Dead viability assay. 0.5% CNT scaffolds showed lower excitation threshold than 0% and 0.1%, which were comparable. Characteristic striation of CM cytoskeletal and contractile proteins was observed in all scaffolds.	CNTs cause increased opacity of polymer solutions which can interfere with polymerisation and mechanical properties of UV-cured materials.	Use of published eStim protocols could further improve CM maturity on CNT-containing scaffolds.
Peña 2017 [69]	Neonatal ventricular cardiomyocytes	1- to 3-day old rat pups	Synthesized MWNTs—carboxylic functionalized	EDC-NHS-lysine cross-linked immobilised CNTs in a PHSU-PNIPAAm-lysine reverse thermal gel.	CM contractions by AFM, IF of α-actinin, vimentin, CD31 and Cx43 plus DAPI, EdU assay, calcium imaging by Fluo-4.	IF showed no CD31-positive cells on any scaffolds but reduced vimentin-positive fibroblasts and increased expression and improved localization of α-actinin and Cx43 in CMs. EdU showed CMs possess more proliferative phenotype after day 3. Calcium imaging and AFM showed increased and synchronised contractions from CMs on CNTs.	No limitations given.	Investigate the biocompatibility and *in vivo* applications of RTG-CNT scaffolds.
Ren 2017 [70]	Neonatal cardiomyocytes	1- to 3-day old Sprague Dawley rat pups	Synthesized MWNTs—6–10 walls, approx. 10 nm diameter, ~200 µm length/Purchased MWNTs—10–30 nm dimeter, 10–30 µm length	Super aligned CNT sheets, produced by CVD, were deposited onto glass or PDMS. Randomly aligned CNT controls were spray deposited onto glass or PDMS.	TUNEL assay, confocal IF of α-actinin and Cx43 plus DAPI, eStim by pacemaker, ImagePro Plus analysis of CM contractility, whole cell patch clamp.	Decreased cell death on SA-CNTs. CMs on SA-CNTs showed increased alignment and elongation. eStim resulted in no changes to cell morphology or viability. Increased expression of Cx43 and improved localization to gap junctions was shown on SA-CNTs. Increased spontaneous and synchronised beating was seen on SA-CNTs, with a decrease in resting membrane potential and action potential duration and an in increase in AP amplitude. Variability in AP duration was also reduced between cells and between consecutive beats on SA-CNTs.	Short CNTs can penetrate the cell membrane and disrupt cell activity.	Investigate biodegradable alternatives to PDMS and the application of scaffolds in cardiac resynchronization therapy.
Roshanbinfar 2017 [71]	HL-1 cardiomyocytes	Murine cell line derived from AT-1 atrial cardiomyocyte tumour lineage	Purchased MWNTs—carboxylic functionalisation modified to carbodihydrazide	Thermoresponsive pericardial tissue-derived hydrogel with CNT concentration of 0.5 wt%.	Alamar Blue assay, Live/Dead assay, confocal IF of α-actinin and Cx43, calcium imaging by Fluo-4.	CMs on CNTs displayed a more proliferative phenotype and increased viability. IF showed increased Cx43 expression but no change in α-actinin. CMs on CNTs showed more synchronous, ‘directional’ beating with BPMs twice as high as controls.	No limitations given.	No further work suggested.
Sherrell 2017 [72]	HL-1 cardiomyocytes	Murine cell line derived from AT-1 atrial cardiomyocyte tumour lineage	Purchased SWNTs—1.5 nm average diameter/Purchased MWNTs—8 nm average diameter	EDC-NHS cross-linked chitosan-collagen hydrogel scaffolds with CNTs at 0.5 or 2 g/L.	Whole cell patch clamp.	SWNT scaffolds had improved mechanical properties compared to MWNTs as a result of increased, even dispersion throughout the hydrogel. Cells on SWNT scaffolds showed similar electrophysiological phenotype to fibronectin/gelatine-coated controls where cells displayed mostly atrial and few pacemaker-type action potentials. Increased in chitosan surfactant in SWNT scaffolds reduced spontaneous beating. Cells on MWNT scaffolds displayed decreased spontaneous beating and increase pacemaker-type action potentials. Ideal scaffold components identified as 2 g/L of SWNTs and 1% chitosan surfactant.	MWNT scaffolds where thought to decrease spontaneous beating due to blockage of potassium channels.	No further work suggested.
Sun 2017 (1) [73]	Neonatal ventricular cardiomyocytes	1-day old Sprague Dawley rat pups	Purchased SWNTs—0.7–1.3 nm diameter, 5–20 µm length, carboxylic functionalisation	GelMA-CNT solutions at 0, 0.5, 1, and 2 mg/mL were dispensed into 24-well plates and cured using UV.	Live/Dead assay, confocal IF of α-actinin, cTnI, Cx43, N-cadherin, PKP2 and PG plus Hoechst 33258, ImageJ analysis of IF staining, video recording of CM beating, TEM, calcium imaging by Fluo-4, WB of β1-integrin, β-catenin, N-cadherin, p-FAK, FAK and RhoA.	High CM viability up to 100 ppm CNT concentration, any higher displayed cytotoxicity. IF showed increased elongation and alignment based on α-actinin with ID proteins all increasing over time. ImageJ analysis also showed increase sarcomere length but also increase in Z-line width. TEM showed increase in ID structures and same observations as ImageJ. Calcium imaging showed stronger synchronous and rhythmic contractions. WB showed β1-integrin increased while β-catenin was unchanged. Blocking β1-integrin caused a reduction in Cx43. Blocking FAK and RhoA also caused this, but further investigation revealed FAK modulates electrical junctions and RhoA regulates mechanical junctions between cells.	No limitations given	Investigate the underlying mechanisms of ID formation on CNTs and the downstream effects on adherens junctions and desmosomes. Also use human cells in scaffolds to optimise modulus for ID formation.
Sun 2017 (2) [74]	Neonatal ventricular cardiomyocytes	1-day old Sprague Dawley rat pups	Purchased SWNTs—0.8–1.6 nm diameter, 5–30 µm length, 95% purity	Type I collagen-CNT solutions, at 0, 0.5, 1, and 2 mg/mL, were combined with DMEMα and FBS and then gelated at 37 °C.	Live/Dead assay, Alamar Blue assay, confocal IF of α-actinin, cTnI and Cx43 plus F-actin and DAPI, TEM, video recording of CM contractions, calcium imaging by Fluo-4, H&E staining.	Experiments revealed dose-dependent toxicity of CNTs based on viability, with 1 wt% being optimum. F-actin staining showed increased cell adhesion and elongation with thicker and longer actin filaments. IF showed an increase in α-actinin, cTnI and Cx43. TEM and H&E also showed increased elongation, alignment, and visibility of sarcomeric structures. Beating analysis showed CMs started to beat earlier and faster on CNTs. Calcium imaging confirmed stronger synchronous and rhythmic contractions on CNTs.	No limitations given.	No further work suggested.
Yu 2017 [76]	Neonatal cardiomyocytes	1-day old Sprague Dawley rat pups	Purchased MWNTs—30 ± 15 nm diameter, 5–20 µm length, 95% purity, carboxylic functionalisation	Type I collagen solutions, in DMEM, with CNT concentrations at 0, 0.5, 1, 2, 5 and 10 wt% were gelated at 37 °C in moulds or well plates.	Video recording of CM contractions.	The number and scaffold coverage of rhythmically contractile CMs was increased on CNT scaffolds.	No limitations given.	No further work suggested.
Liu 2016 [77]	Neonatal cardiomyocytes	1- to 2-day old Sprague Dawley rat pups	Purchased MWNTs—10–20 nm diameter, 10–20 µm length	Aligned blend and coaxial PELA:CNT electrospun fibres at 0, 1, 2, 3, 4, 5, and 6% CNT concentrations.	LDH release assay, SEM, IF of α-actinin and cTnI, ImageJ analysis of CM elongation and aspect ratio, WB for α-actinin and cTnI, custom software analysis of CM contractions.	5% CNT scaffolds showed best cell viability, growth, adhesion, elongation and aspect ratio, highest expression of α-actinin and cTnI by IF and WB, and highest BPM. Coaxial fibres outperformed blend fibres in all parameters. While 5% was optimum, 6% CNT scaffolds began to show cytotoxic effects.	No limitations given.	No further work suggested.
Shin 2015 [78]	Neonatal ventricular cardiomyocytes	2-day old Sprague Dawley rat pups	Synthesized MWNTs—~5 nm inner diameter, ~2 nm wall thickness	CNT forest electrodes were sandwiched between layers of PEG and CNT-GelMA hydrogels.	Confocal IF of α-actinin and Cx43, eStim, custom MATLAB analysis of CM contractions.	Increase in sarcomeric α-actinin organization and alignment but a homogenous distribution of Cx43 which showed variability with changes in CNT concentration and stiffness. CMs showed increased BPM on CNT scaffolds and eStim enabled control over synchronous beating.	No limitations given.	No further work suggested.
Sun 2015 [79]	Neonatal ventricular cardiomyocytes	1-day old Sprague Dawley rat pups	Purchased SWNTs—0.7–1.2 nm diameter, 100–1000 µm length, 95% purity	Type 1 Collagen-CNT solutions, prepared at 0, 0.05, 0.1, 0.15 and 0.2 mg/mL, were deposited onto glass substrates and vacuum dried at 60 °C.	Live/Dead assay, Alamar Blue assay, cell retention by DAPI, qDNA assay, H&E staining, confocal IF of cTnI, α-actinin, Cx43 and N-cadherin plus F-actin and DAPI, TEM, calcium imaging by Fluo-4, WB of Cx43, N-cadherin, plakophilin2, plakoglobin, β-integrin, p-ERK, ERK, p-AKT, AKT, p-Src, p-FAK and ILK, RT-PCR of AP-1, c-fos, MEF-2c, NKX2.5, and GATA4.	Low CNT concentrations showed similar viability to controls, but higher concentrations displayed cytotoxicity. CNT scaffolds showed increase cell retention compared to controls. F-actin staining showed thicker and more spread out actin filaments with better cell adhesion. Increased DNA content across all substrates but no difference between CNTs and controls. H&E and IF staining both showed increased alignment and visibility of sarcomeres with IF showing an increase in TnI, Cx43, and NC. TEM displayed more compact and visible I, Z and H bands and an increase in ID structures. Calcium imaging showed stronger synchronous, rhythmic beating on CNTs. WB showed an increase in all ID proteins except β1-integrin. With an increased in p-FAK but no change in p-Src or ILK. Increase in p-FAK also lead to increases in p-ERK but no change in p-AKT. Blocking ERK lead to decrease in Cx43 while blocking AKT showed an increase as seen in normal culture. In addition, blocking β1-integrin also caused a decrease in Cx43 and N-cadherin but only on CNTs. RT-PCR showed increases in GATA4 and MEF-2c but no change in AP-1, c-fos or NKX2.5.	No limitations given.	No further work suggested.
Kharaziha 2014 [80]	Neonatal ventricular cardiomyocytes	2-day old Sprague Dawley rat pups	Purchased MWNTs—30 ± 15 nm diameter, 5–20 µm length, 95% purity, carboxylic functionalisation	Aligned EDC-NHS cross-linked electrospun PG:CNT fibres with 0.05, 0.5 and 1.5% CNT concentrations.	Live/Dead assay, Alamar Blue assay, cell retention by DAPI, IF of α-actinin, Cx43 and cTnI plus DAPI, ImageJ analysis of IF staining, FFT analysis of cell alignment, custom MATLAB analysis of CM contractions, eStim.	1.5% PG:CNT scaffolds showed the greatest improvement in cell viability, retention, alignment, and contractile function. eStim further improved CM contractile and electrical activity.	No limitations given.	Develop layered and vascularised structures from CNT scaffolds to improve nutrient delivery.
Pok 2014 [81]	Neonatal ventricular cardiomyocytes	1- to 3-day old Sprague Dawley rat pups	Purchased SWNTs—0.98 ± 0.21 nm diameter, 0.1–4 µm length, >90% purity	Chitosan-gelatine hydrogel with 0, 33, 69, 175 ppm CNT concentrations.	Live/Dead assay, SEM, visual analysis of CM contractions, IF of α-actinin and Cx43 plus DAPI, action potential imaging by Di-8-ANEPPS.	Scaffolds containing less than 69 ppm CNTs had 80% viability, but higher concentration showed large drop in viability—indicating CNT toxicity is dose dependent. Analysis of CM BPM showed low CNT concentrations were similar to controls, but medium concentrations displayed BPM close to physiological BPM of rats. IF showed increase in α-actinin and slight increase in Cx43, which was confirmed by WB. Cx43 also showed improved localization to gap junctions. CMs on CNTs showed increased conduction velocity and action potential duration.	Mechanisms of CNT cytotoxicity is unknown and widely debated.	*In vivo* application of scaffolds in a rat disease model.
Tao 2014 [82]	Neonatal cardiomyocytes	1-day old rat pups	Synthesized MWNTs—3–10 walls, 6–15 nm diameter, pristine modified to oxidized by acid vapour	SA-CNT yarns were pulled from aligned CNT arrays to produce thin films of adjacent yarns. 30 films were stacked, rotating alignment 90° each time, to produce scaffolds. CNTs were then oxidised using nitric acid vapour at 120 °C for 6 h.	Cell retention by Hoechst, visual analysis of cell adhesion and growth, SEM, calcium imaging by Fluo-4.	Higher cell count was observed on oxidised CNTs with increased adherence and growth. Conclusions from SEM were unclear. Calcium imaging showed no difference between CNT scaffolds and controls.	Oxidised CNT films had lower cell density than traditional culture dishes.	Oxidised CNTs provide a starting point for functionalisation to create scaffolds specialized for cell adherence.
Martinelli 2013 [84]	Neonatal ventricular cardiomyocytes	1- to 3-day old rat pups	Purchased MWNTs—20–30 nm (parameter not given)	Isotropic CNT layer on glass substrates from 0.01 mg/mL dispersion in DMF, annealed at 350 °C in N₂ for 20 min.	SEM, IF of α-actinin and Cx43, WB of Cx43, SERCA2a and MYH7, qPCR for αMHC, βMHC, ANP, SERCA2a and skeletal actin, pStim by phenylephrine, whole cell patch clamp, calcium imaging by FURA2.	Increased expression of αMHC and SERCA2a, decreased expression of ANP and no change in βMHC or skA. Changes in SERCA2a and myosin heavy chains were also seen. Increased expression of Cx43 and improved localisation to gap junctions. Stimulation by phenylephrine showed CMs on CNTs did not exert the same hypertrophic response, qPCR revealed increased in αMHC and SERCA2a with a decrease in βMHC and skA but no change in ANP. Patch clamp showed decrease in resting membrane potential. Calcium imaging showed increase in more mature, sporadically contractile CMs.	No limitations given.	No further work suggested.
Shin 2013 [85]	Neonatal ventricular cardiomyocytes	2-day old Sprague Dawley rat pups	Purchased MWNTs—30 ± 15 nm diameter, 5–20 µm length, 95% purity, carboxylic functionalisation	GelMA UV-cured hydrogel thin films made with 0, 1, 3, and 5 mg/mL CNT concentrations.	Live/Dead assay, MTS assay, qDNA assay, confocal IF for α-actinin, cTnI and Cx43 plus F-actin and DAPI, WB for α-actinin, cTnI and Cx43, SEM, custom software analysis of CM contractions, eStim, pStim by 1-heptanol and doxorubin.	CNT scaffolds showed increased cell retention and viability with limited proliferation, indicating CMs are growing but fibroblasts are not dividing. Improved alignment of whole cells and cytoskeletal organization. Increased expression of cTnI and increased visibility and organization of sarcomeric structures. An increase in α-actinin and TnI but only a small increase in Cx43 from WB, but IF indicates Cx43 is instead better localized to gap junctions. CMs also showed increased BPM and synchronization. CMs on CNTs showed a reduced threshold potential when undergoing eStim. CNT scaffolds exhibited a protective effect against pStim. SEM revealed CNTs encourage cell elongation and spreading of filopodia.	No limitations given.	Investigation into inhibitor resistance on high concentration CNT scaffolds.
Martinelli 2012 [86]	Neonatal ventricular cardiomyocytes	1- to 3-day old rat pups	Purchased MWNTs—20–30 nm (parameter not given)	Isotropic CNT layer on glass substrates from 0.01 mg/mL dispersion in DMF, annealed at 350 °C in N₂ for 20 min.	TEM, Alamar Blue assay, IF of α-actinin plus DAPI, BrdU assay, flow cytometry and IF of pHH3, whole cell patch clamp.	Frequent and tight contacts existed between cells and CNTs with some CNTs appearing to penetrate membrane. Increased metabolic activity on CNTs after 36 hrs. IF of α-actinin revealed fewer α-actinin-negative fibroblasts on CNTs. BrdU, pHH3 and flow cytometry all confirmed CMs on CNTs retained a more proliferative phenotype. Patch clamp showed lower resting potential and shorter action potential duration on CNTs.	The precise effects of MWNTs are not understood.	Investigation into underlying effects with regards to genotype and phenotype, and into applications treatment of arrhythmia and conduction diseases.

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
