# Peer review of "Carbon Nanotube-Based Scaffolds for Cardiac Tissue Engineering—Systematic Review and Narrative Synthesis"

_bioengineering, 2021, doi:10.3390/bioengineering8060080_

Round 1

Reviewer 1 Report

This is the reviewer’s comment on the review article titled “Carbon Nanotube-Based Scaffolds for Cardiac Tissue Engineering – Systematic Review and Narrative Synthesis” authored by Scott L et al. This review following PRIMA guidelines evaluates the use and efficacy of carbon nanotubes for cardiac tissue scaffolds. This is an interesting and a well written review. The authors had performed extensive literature survey and complied them here. Kudos to the authors for making an extensive and comprehensive Table 2. This review can be accepted for publication after the following minor comments were addressed.

The second and the third sentences of the abstract needs to be changed to a more meaningful sentence. [Tissue engineering is a solution to this crisis enabling the development of biomaterials which facilitate more dynamic and complex in vitro models. A material that has drawn attention are carbon nanotubes (CNTs).]

Though it's a well written article, the English needs to be improved throughout the article.

Examples:

  1. Limitations of this study. “It is important acknowledge” change to “It is important to acknowledge”

Typo “Completing interests”

Author Response

This is the reviewer’s comment on the review article titled “Carbon Nanotube-Based Scaffolds for Cardiac Tissue Engineering – Systematic Review and Narrative Synthesis” authored by Scott L et al. This review following PRIMA guidelines evaluates the use and efficacy of carbon nanotubes for cardiac tissue scaffolds. This is an interesting and a well written review. The authors had performed extensive literature survey and complied them here. Kudos to the authors for making an extensive and comprehensive Table 2. This review can be accepted for publication after the following minor comments were addressed.

Thank you

The second and the third sentences of the abstract needs to be changed to a more meaningful sentence. [Tissue engineering is a solution to this crisis enabling the development of biomaterials which facilitate more dynamic and complex in vitro models. A material that has drawn attention are carbon nanotubes (CNTs).]

These sentences have been combined and now read as follows “Tissue engineering is a solution to this crisis and in combination with the use of carbon nanotubes (CNTs), which have drawn recent attention as a biomaterial, could facilitate the development of more dynamic and complex in vitro models.” We hope this satisfies your request for a more impactful statement.

Though it's a well written article, the English needs to be improved throughout the article.

Examples:

  • Limitations of this study. “It is important acknowledge” change to “It is important to acknowledge”
  • Typo “Completing interests”

The above corrections have been implemented and the manuscript has been proofread to remove all other mistakes in spelling and grammar.

Reviewer 2 Report

I like the systematic approach of this review and the detailed description of the methods. The discussion is interesting to read and provides a good overview on the field.  There is lots of information provided. However, the advantage of CNT enriched scaffolds about other types of non-conducting or conducting (for example graphene oxide loaded scaffolds) remains vague.

Points to be considered for the resubmission:

  • (Introduction) line 59. It is not perfectly correct to state that culture cardiomyocytes fail to emulate the behavior of cardiac muscle in vivo because novel culturing methods have evolved recently. Please check the following report:Cell Stem Cell. 2020 Jun 4;26(6):862-879.e11. doi: 10.1016/j.stem.2020.05.004. Epub 2020 May 26.

  • (Introduction) line 62: I don´t agree with the authors that there is no suitable matrix for cardiomyocyte culture. Especially polyacrylamide matrices with tissue matched elasticity have been used successfully to culture iPS cell derived cardiomyocytes for extended periods supporting their function. It could help to distinguish between 2D and 3D culture models.

  • (Introduction) line 102: I don´t agree that cardiomyocytes require electrically conductive scaffolds. These cells form a functional syncytium via connexin 43 gap junctions and electrical excitations are spread via connexins. It can easily be observed that iPS cell derived cardiomyocytes form synchronously contracting cell layers on non-conduction surfaces.

  • Figure 1: there must be a typo: If you have 76 papers and 46 fail to match criteria there are 30 papers left and not 76.

  • It´s quite unusual to neglect the work of colleagues who are not publishing in open access journals (non-open access as exclusion criterion). These groups may not be able to afford the open access fee but want to bring their research into a recognized journal. At most universities it is a must to publish in high impact journals to have a chance for funding and for a career in science and this consideration is more important to scientists than open access options. Of course this is highly debatable but it won´t help to exclude scientific work from a review because it is not open access. Authors should justify this please.

Author Response

I like the systematic approach of this review and the detailed description of the methods. The discussion is interesting to read and provides a good overview on the field.  There is lots of information provided. However, the advantage of CNT enriched scaffolds about other types of non-conducting or conducting (for example graphene oxide loaded scaffolds) remains vague.

Thank you. It was the aim of this review to assess the impact of CNTs on cardiac tissue engineering over the past decade, which does not necessarily include endorsing or comparing their use to other materials, but instead highlighting them as an incredibly useful but underutilised tool for researchers to take advantage of. Your specific suggestions and corrections to the manuscript have been addressed below.

Points to be considered for the resubmission:

(Introduction) line 59. It is not perfectly correct to state that culture cardiomyocytes fail to emulate the behavior of cardiac muscle in vivo because novel culturing methods have evolved recently. Please check the following report:Cell Stem Cell. 2020 Jun 4;26(6):862-879.e11. doi: 10.1016/j.stem.2020.05.004. Epub 2020 May 26.

Thank you. We have changed this statement to acknowledge that there are instances in which cultured CMs are representative their native state in the myocardium, particularly when multi-cell systems are utilised, and have included your suggested citation to reference this.

(Introduction) line 62: I don´t agree with the authors that there is no suitable matrix for cardiomyocyte culture. Especially polyacrylamide matrices with tissue matched elasticity have been used successfully to culture iPS cell derived cardiomyocytes for extended periods supporting their function. It could help to distinguish between 2D and 3D culture models.

Thank you for your feedback. We have now clarified that from our research, few tissue scaffolds satisfy the full spectrum of stimuli required by cardiomyocytes with the implication that CNTs can be used to create platforms which improve mechanical and electrical properties of cardiac tissue scaffolds.

(Introduction) line 102: I don´t agree that cardiomyocytes require electrically conductive scaffolds. These cells form a functional syncytium via connexin 43 gap junctions and electrical excitations are spread via connexins. It can easily be observed that iPS cell derived cardiomyocytes form synchronously contracting cell layers on non-conduction surfaces.

Thank you for your comment. We agree that it is obvious that cardiomyocytes are capable of forming synchronously contracting sheets on their own. We intended for this statement to address electrical stimulation as an important feature of cardiac tissue engineering to encourage desirable cell behaviours, and how the application of CNTs as conductive scaffolds permits the use of this kind of stimulation. We have now altered this. We were also able to show from our review of the literature that just the presence of CNTs or increased scaffold conductivity, even without electrical stimulation, does have positive effects on cardiomyocyte culture compared to non-conductive controls – which is mentioned in the Discussion of the manuscript.

Figure 1: there must be a typo: If you have 76 papers and 46 fail to match criteria there are 30 papers left and not 76.

Thank you, this error has been amended.

It´s quite unusual to neglect the work of colleagues who are not publishing in open access journals (non-open access as exclusion criterion). These groups may not be able to afford the open access fee but want to bring their research into a recognized journal. At most universities it is a must to publish in high impact journals to have a chance for funding and for a career in science and this consideration is more important to scientists than open access options. Of course, this is highly debatable but it won´t help to exclude scientific work from a review because it is not open access. Authors should justify this please.

Thank you for your feedback on this. We ensured that we made every effort to obtain full texts of the articles which were not Open Access. As described in our Methods section, “if the full publications were not available through open access, the authors were contacted via the paper’s correspondence email. Failing that the paper was excluded from the final review set.” This is a standard practice for systematic reviews and the consequences of this choice are also addressed in the Limitations of the Study section of the manuscript.

Reviewer 3 Report

  • Make sure to define all acronyms in the text, e.g., "PRISMA" needs definition in the Abstract.
  • Table 1 must be presented in more informative and simpler way. The current form is too confusing and hard to follow. I suggest consolidating or removing some of the columns, and using a different symbol, other than the check marks, for the table cells. Maybe a graph could better present these data?

Author Response

Make sure to define all acronyms in the text, e.g., "PRISMA" needs definition in the Abstract.

Thank you for bringing this to our attention. PRISMA has now been defined where it has been used in the Abstract.

Table 1 must be presented in more informative and simpler way. The current form is too confusing and hard to follow. I suggest consolidating or removing some of the columns, and using a different symbol, other than the check marks, for the table cells. Maybe a graph could better present these data?

Thank you for your comment. We have reformatted the table so column titles are more succinct, with different criteria titles defined in the methods section and table legend, and borders between all cells have been coloured black so the results of each publication’s appraisal are clear. We will ensure that both tables are formatted in landscape for final publication so their contents are easy to interpret for readers.

Round 2

Reviewer 2 Report

All my concerns were addresses appropriatly. I recommend acceptance of the manuscript.